# Numerical Investigation on the Effect of Blockage on the Icing of Airfoils

**Daixiao Lu** [1,2] **, Zhiliang Lu** [1,]***, Zhirong Han** [2]**, Xian Xu** [2] **and Ying Huang** [2]

1 Key Laboratory of Unsteady Aerodynamics and Flow Control, Ministry of Industry and Information Technology, Nanjing University of Aeronautics and Astronautics, Nanjing 210016, China
2 Shanghai Aircraft Design and Research Institute, Shanghai 201210, China
* Correspondence: luzl@nuaa.edu.cn

**Abstract:** The blockage is one of the important factors affecting the icing of airfoils in wind tunnel tests. In this paper, numerical simulations are conducted to study the effect of blockage on the icing of different airfoils. By reducing the height of testing wind tunnels, the blockage is increased, and the changes in the height and angle of the ice horn are numerically investigated. The simulation results indicate that as the blockage increases, the flow velocity above the stagnation point of the airfoil increases, leading to larger pressure coefficients distribution and stronger heat transfer capacity. As a result, the position of icing moves forward, and the angle of the upper ice horn becomes smaller. In addition, the increased flow velocity facilitates the collection of water droplets in the area, which improves the icing and increases the height of the upper ice horn. It is also found that the blockage increases the angle of attack of the airfoil, moving the stagnation point backward and decreasing the angle of the upper ice horn. When the blockage is above 15%, the joint influence of the opening angle and height of the upper ice horn significantly reduces the projection height of the upper ice horn in the direction of the incoming flow, leading to unacceptable criticality of the ice shape.

**Keywords:** blockage; icing of airfoils; stagnation point; angle of attack; criticality of the ice shape

## 1. Introduction

Ice accretion occurs immediately on wing, tail, propeller, rotor and even antenna when aircraft fly through the cloud full of supercooled water droplets or encountering precipitations such as freezing rain or drizzle. Ice accumulating on the aerodynamic sensitive surfaces leads safety issue because of its detrimental effect on the performance such as maximum lift penalty, stall angle reduction, and parasite drag increase. Aviation accident data showed that 583 icing accidents occurred and caused more than 800 fatalities from 1982 to 2000 in the United States [1]. For this reason, aircraft icing was even recognized as the "most wanted aviation transportation safety improvement" by the National Transportation Safety Board (NTSB) [2]. The Federal Aviation Regulations (FAR) Part 25 Appendix C define icing envelopes (in terms of air temperature, liquid water content and droplets' size) for aircraft certification, corresponding to 99.9% of icing conditions found in stratiform clouds [3]. Most light aircraft are not required to pass this certification and do not usually have the required ice-protection systems for flying into icing conditions. Larger aircraft are equipped with a variety of anti-icing or de-icing systems (heaters, pneumatic boots and liquid flows) to help them prevent ice formation [4].

In order to guarantee safe and on schedule operations of transportation aircraft, accurate prediction of ice accretion, related performance degradation and anti/de-icing systems development play an important role for the design of aircraft. Over the years, test approaches including flight and wind-tunnel tests, and numerical simulations have been adopted in the prediction of ice accretion. Following the development and progress of computational science as well as the great advantages of numerical simulation in the economic cost, many ice-prediction codes have been successfully developed and serve as valuable

design and certification tools for the aircraft manufacturers such as LEWICE developed by NASA Lewis Research Center [5], ONERA 2-D/3-D developed by ONERA [6], IMPIN3D developed by Italy [7], and the second-generation 3-D icing simulation system FENSAP-ICE developed by Numerical Technologies International [8], etc. Recently, a large number of articles have been published that are devoted to the application of machine learning methods, deep neural networks in the field of fluid dynamics and even the procedure and method for the ice accretion prediction for different airfoils using artificial neural networks (ANNs) are discussed [9]. It is worth expecting that the combination of two approaches: machine learning and numerical simulation, will speed up the prediction of the icing shape and significantly reduce computational costs.

Considering that the natural icing conditions are very difficult and costly to obtain, the icing test in wind tunnels is not only an important means to study aircraft icing (an essential step to obtaining civil aircraft airworthiness certification) [10,11] but also an important method to develop and verify numerical tools used for icing [11–14]. The civil aircraft authorities have set clear approval requirements for the wind tunnels and numerical tools used for icing in the design of aircraft [15,16]. Previous wind tunnel tests show that wall interference is an important factor affecting the accuracy of testing results [17–19]. In a closed wind tunnel, the existence of walls limits the bending and diffusion of streamlines. Therefore, the area through which the airflow between adjacent streamlines is smaller than that in the free space, and the average flow velocity between the airfoil and tunnel wall is larger. This phenomenon is called the blocking effect. In reality, the limitation of the bending of streamlines is equivalent to inducing an upward angle to the airfoil, thereby increasing the angle of attack as compared with that defined using the axis of the wind tunnel [18]. The increased angle of attack will move the icing from the upper surface of the airfoil to the lower surface, leading to overestimated aerodynamic performances of airfoils. Consequently, the criticality of the ice shape is not guaranteed, and the test results will not be recognized.

The blockage is defined as the ratio of the maximum incoming airflow area to the cross-sectional area of the test tunnel. The US Federal Aviation Administration [20] reported that the blockage should not exceed 10% in the icing test of airfoils in wind tunnels. However, in practice, the chord length of the wings of large passenger aircraft is generally long (may exceed 5 m). Even the world's largest wind tunnel (the FL-16 ice wind tunnel [21] of the China Aerodynamic Research and Development Center, with a size of 4.8 m × 3.2 m) is not suitable for such experiments. On the other hand, the aircraft airworthiness examiners generally do not allow the use of scaled test models in icing wind tunnel tests. Even though the hybrid wing design technology is adopted [22,23], the blockage will still exceed 10%, which makes the reliability of icing test results questionable under certain circumstances (e.g., with a large angle of attack).

Guo Qiling et al. [24,25] of the Key Laboratory of Icing and Deicing of CARDC studied the influence of walls on the impinging characteristics of water droplets and icing of airfoils. Their research shows that the blockage will affect the impinging characteristics of water droplets, and the increased blockage will improve icing on the surface of airfoils. Zocca et al. [26] found that the blockage has a significant impact on the icing of airfoils and brings a large deviation to the ice shape. Qin et al. [27] numerically examined the influence of cavity walls on the impinging characteristics of water droplets. They concluded that the blockage would shift the stagnation point of airflows on the airfoil, which significantly influences the collection of water droplets on the airfoil surface. The NRC wind tunnel in Canada also noticed that the increased blockage is beneficial for the collection of water droplets. They revealed the relationships between the water droplet collection rate and the Reynolds number of water droplets at different blockages [28]. It can be seen from the studies mentioned above that the impinging characteristics of water droplets and resultant ice shape are both affected by the blockage. Nonetheless, there is still no recommended value of the maximum blockage acceptable in wind tunnel tests. In this paper, the NACA0012 and GLC305 airfoils in the literature are used as examples and numerically investigated.

By adjusting the height of the testing wind tunnel, the icing characteristics of airfoils at different blockages are simulated. Based on the ice shapes obtained from the tests, the influence of the blockage on the ice shape is studied, and the maximum blockage allowed in the icing tests of airfoils is explored. The outline of this article is organized as follows. Section 2 introduces the numerical methodology employed in this research. Section 3 validates the numerical model. Section 4 presents and discusses the simulation results. Finally, concluding remarks are drawn in Section 5.

## 2. Numerical Methodology

### 2.1. Model Description and Mesh Configuration

Figure 1a presents the sketch of the airfoil in a wind tunnel under investigation in this study. In the numerical simulations, for simplicity, the airfoil is installed without fixed boundary conditions, and its left and right supporting structures are omitted. The blockage is changed by adjusting the height between the upper and lower tunnel walls, ignoring the interference of the side tunnel walls. For instance, If the height of the teste section is decrease and the installation angle of the wing model remains the same, the blockage is increased. As shown in Figure 1a, two-dimensional unstructured grids are adopted to mesh the airfoil and the surrounding flow field. To save computational cost, denser grids are constructed in the airfoil region as compared with the surrounding flow field that employs relatively coarse grids. As such, triangular prism boundary layer grids are used on the surface of the airfoil, as displayed in Figure 1b. Likewise, triangular pyramid grids are employed near the leading edge of the airfoil, as shown in Figure 1c. The prism layers are 40, the first layer is about $1 \times 10^{-6}$ m high, and the expansion ratio is 1.2, the number of grid cells along the airfoil is 120.

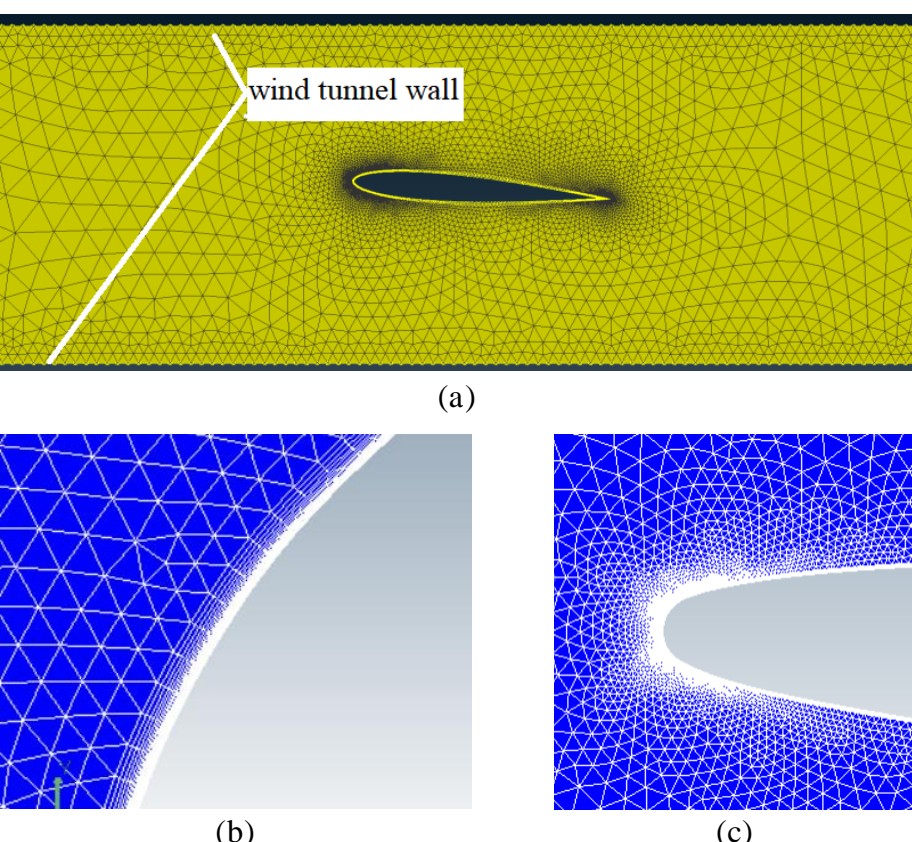

**Figure 1.** Model description and mesh configuration. (**a**) Sketch of the airfoil in a wind tunnel. (**b**) Boundary layer grids. (**c**) Grids near the edge of the airfoil.

### 2.2. Governing Equations

Following the construction of mesh for the airfoil in the wind tunnel, numerical calculations are conducted to simulate the ice accretion. The working fluid is air and follows the Navier-Stokes equations and the equation of state, which are,

$$\frac{\partial \overline{\rho}}{\partial t} + \frac{\partial}{\partial x_i}(\overline{\rho} \widetilde{u}_i) = 0 \tag{1}$$

$$\frac{\partial}{\partial t}(\overline{\rho} \widetilde{u}_i) + \frac{\partial}{\partial x_j}\left(\overline{\rho} \widetilde{u}_i \widetilde{u}_j + \overline{p} \delta_{ij} - \widetilde{\tau_{ij}^{tot}}\right) = 0 \tag{2}$$

$$\frac{\partial}{\partial t}(\overline{\rho} \widetilde{e}_{tot}) + \frac{\partial}{\partial x_j}\left(\overline{\rho} \widetilde{u}_j \widetilde{e}_{tot} + \widetilde{u}_j \overline{p} + \widetilde{q_j^{tot}} - \widetilde{u}_i \widetilde{\tau_{ij}^{tot}}\right) = 0 \tag{3}$$

$$p = \rho R_g T \tag{4}$$

where the overbars "~" and "−" denote the density-weighted time averaging (Favre averaging) and classical time averaging (Reynolds averaging), respectively. In this case, $p$, $\rho$ and $T$ are pressure, density and temperature, $u_i$ or $u_j$ stands for the velocity component in corresponding direction, $e_{tot} = e + u_i u_i /2$ is the total energy with $e$ being the specific internal energy, $\tau_{ij}^{tot}$ is the total stress tensor, $q_j^{tot}$ is the heat flux, $\delta_{ij}$ is the Kronecker delta, and $R_g$ is the specific gas constant. The Spalart–Allmaras turbulence model is used in the flow field calculation. In the calculation of droplet trajectory, a single average diameter called Median Volumetric Diameter (MVD) is used, and the coupled heat transfer calculation related to the anti-ice system is not involved.

The governing equations stated above are resolved numerically using the commercial CFD (computational fluid dynamics) package ANSYS Fensap-ice R19.0. Fensap-ice R19.0 includes many modules, such as flow field calculation, droplet collision calculation, ice accretion, and heat transfer calculation, which can be used for icing analysis.

### 2.3. Boundary Conditions

The boundary conditions in the numerical simulations are defined referring to the NACA0012 and GLC305 airfoils. Table 1 lists the parameters of NACA0012 and GLC305 airfoils used in the icing test in NASA Lewis icing Research Tunnel (IRT) [13].

**Table 1.** Parameters of airfoils used in the numerical simulations.

|  | NACA0012 Airfoil | GLC305 Airfoil |
|---|---|---|
| Chord length | 21 in | 36 in |
| Velocity | 102.8 m/s | 90 m/s |
| Angle of attack | 4° | 4.5° |
| Time | 7 min | 16.7 min |
| LWC | 0.55 g/m$^3$ | 0.405 g/m$^3$ |
| MVD | 20 μm | 20 μm |
| Temperature | 265.37 K | 267.40 K |

### 2.4. Sensitivity Studies

Sensitivity studies on the influence of grid number and icing step number on the simulation results are performed. In the multistep icing calculation, the multi-step method based on artificial mesh reconstruction is adopted to ensure that the reconstructed mesh can accurately describe the ice shape and maintain sufficient mesh quality. The time interval of each step of mesh reconstruction is equal: the icing time of each step is defined as the total icing time divided by mesh reconstruction steps.

Figure 2 shows the ice shapes calculated at different grid numbers and mesh reconstruction steps. The "160k 6steps" in the figure represents the case in which the total grid number is 160,000 and the icing step number is 6. First, by comparing the ice shapes at grid numbers of 160,000, 300,000 and 476,000, It has been found that the grid number

of 476,000 is sufficient enough to accurately describe the ice shape while minimizing the computational cost simultaneously. Therefore, the optimal grid number is chosen at 476,000. Subsequently, by comparing the ice shapes at icing step numbers of 3, 6 and 7 when the grid number is 476,000, It has been noticed that the mesh reconstruction time has a great impact on the ice horn height. When the icing step number reaches 6, the ice shape becomes less sensitive. Hence, 6 steps of icing are adopted in the subsequent simulations in this paper.

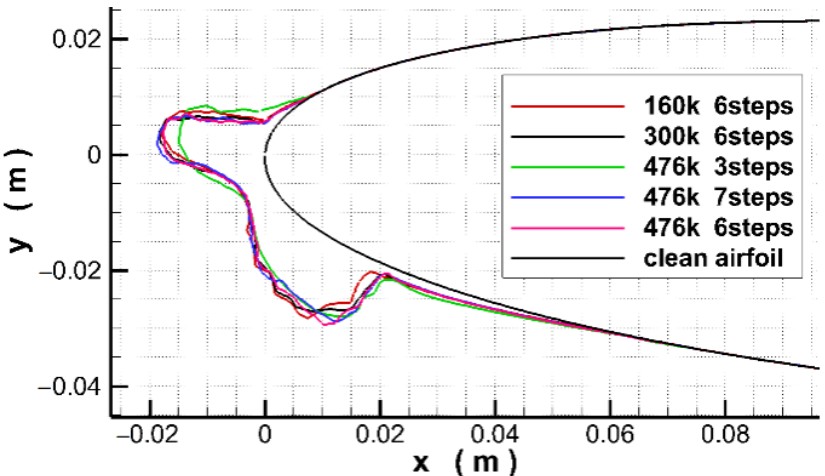

**Figure 2.** Calculated ice shapes at different grid numbers and icing steps.

## 3. Verification of Numerical Results

Comparisons are made between the experimental data of the NACA0012 airfoil in the literature [13] and the ice shape calculated by ANSYS Fensap-ice R19.0 using the same parameters. The results in Figure 3 indicate that the criticality of ice shape from numerical simulations in terms of ice horn height and angle is consistent with that of the experiment, which demonstrates the validity of the numerical methodologies proposed in this research.

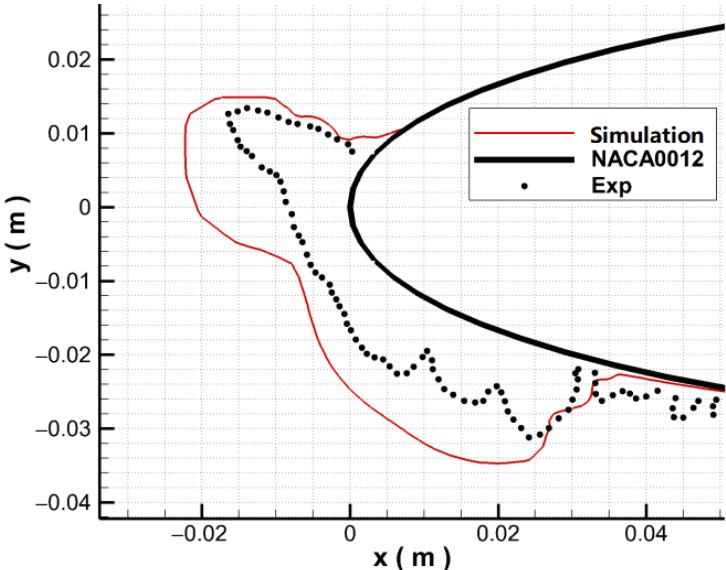

**Figure 3.** Comparison of ice shapes from the experiment and numerical simulations.

## 4. Results and Discussion

### 4.1. Criticality of Ice Shape

Gray [29] demonstrated that the height, position, and angle of the upper ice horn are the critical parameters that affect the formation of ice shape which in return influences the aerodynamic performance of the airfoil. The angle of the upper ice horn has a direct impact on the size and position of bubbles separating on the surface of the airfoil while the parameters of the lower ice horn generally affect the range of negative angle of attack, having little influence on the aerodynamic characteristics of the airfoil. Figure 4a displays the height of the upper ice horn $h$, the chord length of the upper ice horn $c$, and the opening angle of the upper ice horn $\theta$. The characteristic position $x$ of the ice shape is determined by the intersection of the trailing edge of the ice horn and the surface of the airfoil. The characteristic height $h$ is the distance between the intersection described above and the highest point of the ice horn. In practical work, the projection height $H$ of the upper ice horn (see Figure 4b) in the incoming flow direction is generally used to judge the criticality of the ice shape. Specifically, the larger $H$ is, the more critical the ice shape is [30]. The parameter $H$ reflects the joint influence of $h$, $x$, and $\theta$.

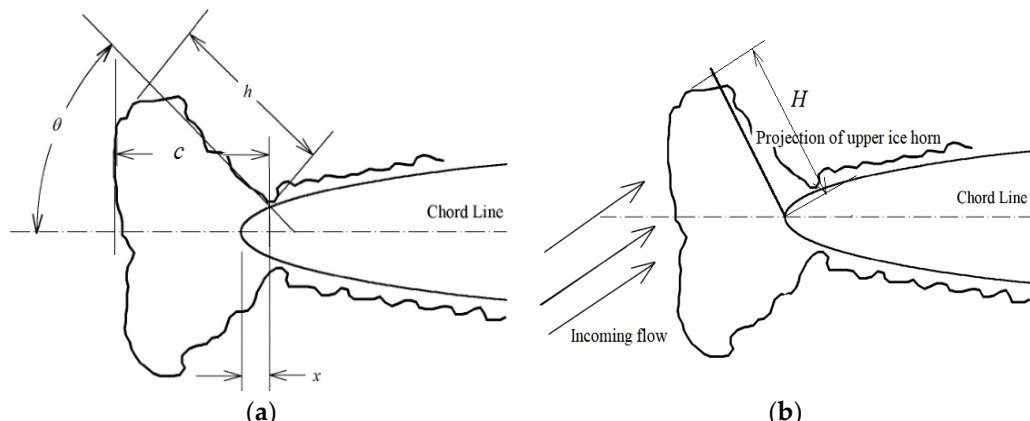

**Figure 4.** Characteristic parameters of the ice horn. (**a**) Ice horn dimensions. (**b**) Projection of upper ice horn in the direction of the incoming flow.

### 4.2. Effect of Blockage on the Criticality of Ice Shape

To simulate the installation of supports (left and right) of the airfoil in practical work, the upper and lower tunnel walls are moved toward the airfoil, thus increasing the blockage. Three blockages (10%, 15.2%, and 20%) are investigated in the simulations and the pressure coefficient distribution of each blockage is calculated. The results show that when the angle of attack is unchanged, as the blockage increases, the peak of the pressure coefficient distribution increases (see Figure 5a), and the stagnation point moves backward (see Figure 5b). This phenomenon is also reported in reference [17] which concludes that the average flow velocity between the airfoil and the tunnel wall increases after the blockage increases, leading to more restrictions on the streamline bending by the tunnel wall. Thus, the angle of attack increases because of the increase of the angle (upward) of the incoming flow.

In the simulations, the ice shapes at different blockages are calculated and presented in Figure 6. By comparing the calculated ice shape without a fixed boundary with that measured in the experiment, it has been found that as the blockage increases, the height of the upper ice horn increases, and the opening angle of the upper ice horn decreases. The increase in the height of the upper ice corner is due to the increase of local velocity around the head of the airfoil, thereby facilitating the collection of water droplets.

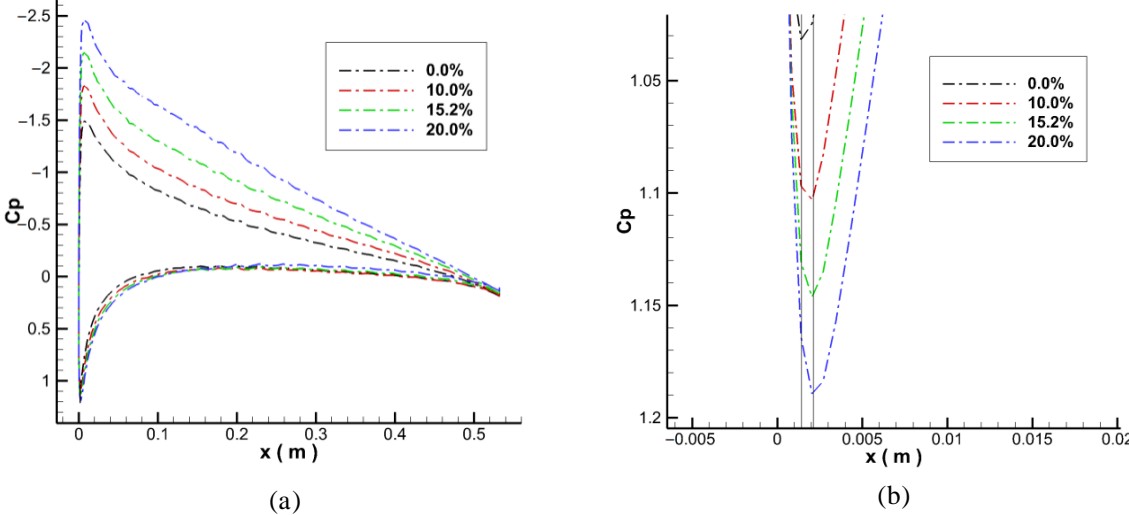

**Figure 5.** (**a**) Pressure coefficients distribution at different blockages. (**b**) Positions of stagnation point at different blockages.

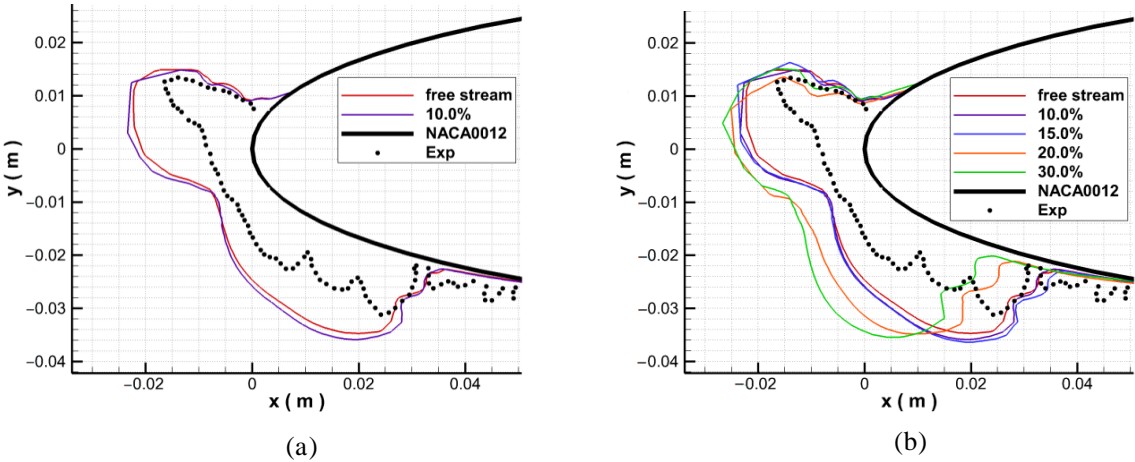

**Figure 6.** Ice shapes at different blockages. (**a**) Ice shape with 10% blockage. (**b**) Ice shape of NACA0012 airfoil at different blockages.

The local droplet collection efficiency β is defined as the ratio, for a given mass of water, of the area of impingement to the area through which the water passes at some distance upstream of the airfoil. Taking a unit width as one dimension of both area terms, the local droplet collection efficiency can then be defined as,

$$\beta = \frac{dy}{ds} = \frac{\Delta y_0}{\Delta s} \tag{5}$$

where $\Delta y_0$ is the spacing between water droplets at the release plane and $\Delta s$ is the distance along the airfoil surface between the impact locations of the same two droplets.

As can be seen from Figure 7, the local droplet collection coefficient β increases with the increase of blockage, and the peak of β gradually moves towards the leading edge. Meanwhile, due to the increase of flow velocity in this area, the heat transfer rate is enhanced which results in faster icing of water film, smaller flow distance, and reduced angle of the upper ice horn. Figure 6b shows that when the blockage reaches 20% or even 30%, the change in the opening angle of the upper ice horn becomes more obvious.

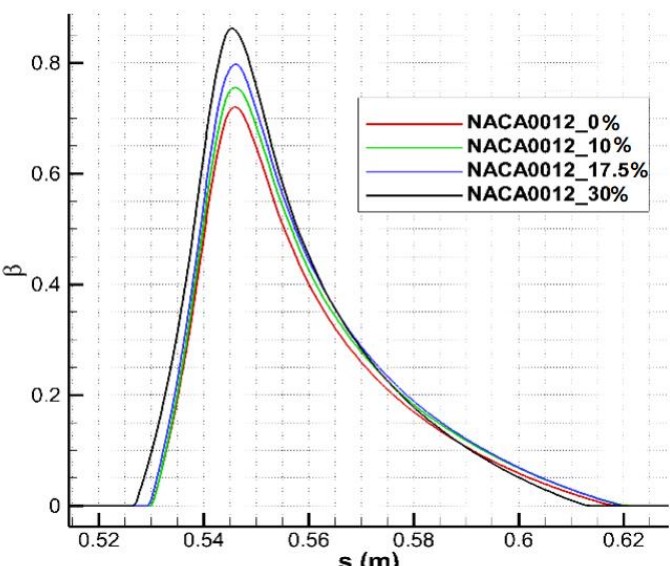

**Figure 7.** Droplet collection coefficients at different blockages of NACA0012 airfoil.

Numerical simulations are also performed on the airfoil of GLC305 reported in the literature [13]. As can be seen in Figure 8, as the blockage increases, the opening angle of the upper ice horn decreases. In particular, when the blockage increases to 17.6%, there is an obvious decline in the opening angle of the upper ice horn. Figure 9 shows the relationship between the blockage and the dimensionless projection height H/c of the upper ice horn in the direction of incoming flow, where *c* represents the chord length of the airfoil. It is found that *H/c* is small when the blockage falls between 10% and 15%. When the blockage is greater than 15%, *H/c* decreases remarkably, which significantly affects the criticality of the ice shape.

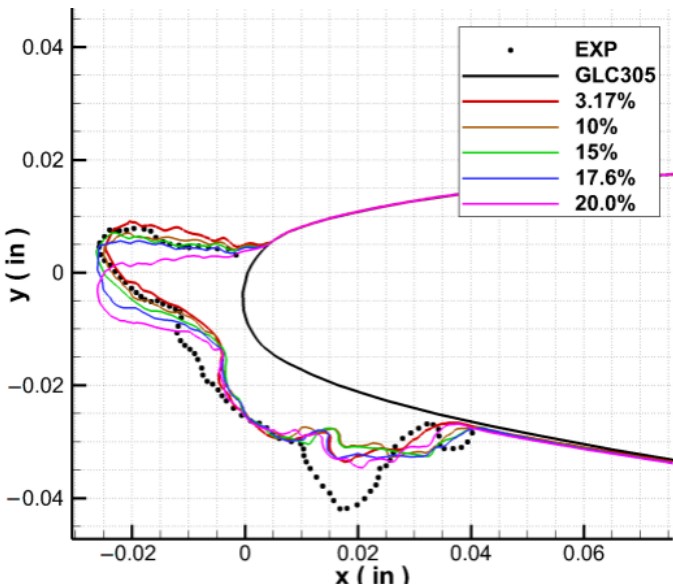

**Figure 8.** Ice shape of GLC305 airfoil at different blockages.

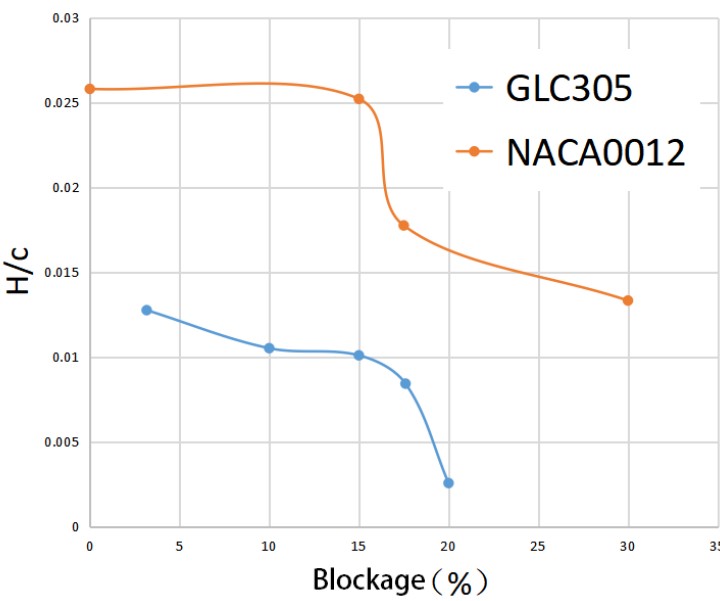

**Figure 9.** Relationship between the blockage and the dimensionless projection height.

## 5. Conclusions

This study investigates the effect of blockage on the icing of airfoils through computational fluid dynamics. Simulations are performed on the airfoils of NACA0012 and GLC305 tested in the wind tunnels, and comparisons are made between the calculated results and experimental data in the literature. The criticality of ice shape from numerical simulations in terms of ice horn height and angle is consistent with that of the experiment, which demonstrates the validity of the numerical methodologies proposed in this study. The key conclusions of this research are summarized as follows.

(1) The increase of ice horn height will increase the criticality of ice shape, while the decrease of ice horn angle will reduce the criticality of ice shape

(2) As the blockage increases, the peak of the pressure coefficient distribution increases, and the stagnation point moves backward. This is because the average flow velocity between the airfoil and the tunnel wall increases after the blockage increases, leading to more restrictions of the streamline bending by the tunnel wall.

(3) With the increase of the blockage of the ice tunnel, the joint influence of the opening angle and height of the upper ice horn significantly reduces the projection height of the upper ice horn in the direction of the incoming flow. As a result, the criticality of the ice shape is reduced. To ensure the criticality of the ice shape, the blockage should be below 15%.

**Author Contributions:** Conceptualization and methodology, Z.L. and D.L.; methodology and software Z.H.; validation, X.X. and Y.H.; formal analysis and investigation, D.L.; resources and data curation, Z.H.; writing—original draft preparation, D.L.; writing—review and editing, Z.L.; visualization, Z.H.; supervision, Z.L.; project administration, D.L.; funding acquisition, Z.L. All authors have read and agreed to the published version of the manuscript.

**Funding:** This research was funded by the National Natural Science Fundation of China (grant nos. 12102187 and 11872212), the National Key R&D Program of China (Project No. 2020YFA0712000), and a project funded by the Priority Academic Program Development of Jiangsu Higher Education Institutions.

**Conflicts of Interest:** The authors declare that they have no known competing financial interest or personal relationships that could have appeared to influence the work reported in this paper.

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
