# Peer review of "Numerical Investigation on the Effect of Blockage on the Icing of Airfoils"

_aerospace, doi:10.3390/aerospace9100587_

Round 1

Reviewer 1 Report

The authors investigate the influence of blockage on the ice accretion on airfoils. It is a highly relevant topic as the icing tests are necessary for regulations and there is no current guidelines on blockage. This study is a useful contribution in this regard. I recommend the study for publication considering the authors fixes the minor issues mentioned in the attached annotated PDF.

Author Response

Thank you for your comments. Those comments are valuable and very helpful for improving the quality of the manuscript. We have read through the comments carefully and modified the manuscript in response to the comments. Now I provide a point-to-point response to the reviewers’ comments in the attachment and highlight the changes in the revised manuscript. We sincerely hope that you are satisfied with our responses and modifications.

Reviewer 2 Report

1.       Introduction section of the manuscript can be improved with more appropriate research papers.

2.       Logical arrangement of the literature review also can be improvised.

3.       Justify with reference: “aircraft airworthiness examiners generally do not allow the use of scaled test models in icing wind tunnel tests”. Is this the author(s) claim statement or is it mentioned somewhere in any paper? Please clarify.

4.       Paragraph splitting needs care. The last paragraph of the Introduction section was the continuation of the previous paragraph yet the author(s) have split them into a new paragraph. Care must be taken while splitting/forming a new paragraph. It is advised for the author(s) to revise the same.

5.       The following sentence needs clarity – advised to rephrase to improve the quality of the information provided –“Nonetheless, there is still no recommended value of the maximum blockage acceptable in wind tunnel tests.”

6.       The author(s) have altered the height between the upper and lower tunnel walls, ignoring the interference of the side walls. Since the blockage is calculated based on the area of the test section to the frontal area of the model. Kindly please clarify – ignoring the side walls does not influence your experiments on the blockage.

7.       Methodology of testing is not adequately explained. Need more details. A schematic representation or the inclusion of the actual image representing how the different blockages were simulated can be added for better understanding.

8.       In section 2.3 Boundary conditions, the details of the parameters mentioned in writing can be removed or table 1 can be removed avoiding duplication.

9.       In section 2.4, the use of words like “We” can be avoided and academic writing will be much appreciated. Rephrase “We” by” It has been found that” etc. as preferred by the author(s).

10.   Why step number 3 has been performed only for 476k grid alone and not for 160k or 300k grids?

11.   Mention the % of deviation for validation studies performed between the original literature and the numerical simulations.

12.   Author(s) have noted down some interesting phenomenon like “when the blockage increases, angle of attack increases because of the increase of the angle (upward) of the incoming flow”. The reviewer appreciates the author(s) for noting such important phenomena. However, the addition of flow contours/flow field structures will further improve the understanding of the reader.

13.   In section 4.1, the opening angle of horn ice can also be mentioned for enhanced understanding to the readers. Additionally, this will supplement the discussions about the opening angle in section 4.2 for both the NACA and GLC airfoils.

14.  Figure 7 and droplet collection efficiency () is not adequately explained by the author(s). Elaborate the findings in detail.

15.   All the results are quantitatively described and this paper presents results as mere data and the underlying flow physics / qualitative flow characteristics lack investigation. The author (s) should take more effort into describing the flow physics in addition to this quantitative information.

16.   Editor appreciates the findings relating H/c to the blockage. More detailed description will be much appreciated.

Author Response

(The authors gave the same response as above.)

Reviewer 3 Report

Review of a manuscript titled " Numerical investigation on the effect of blockage on the icing of airfoils" 

The authors have tried studying the effect of the wind tunnel blockage on icing for flow over an airfoil using numerical simulations. The aspects listed below must be considered before the manuscript can be accepted. 

1. The literature review is too narrow as the problem addressed is also very specific. It would be better for authors to enhance the view of the literature on generic icing simulations methods and progress a bit first like

Strijhak S, Ryazanov D, Koshelev K, Ivanov A. Neural Network Prediction for Ice Shapes on Airfoils Using iceFoam Simulations. Aerospace. 2022; 9(2):96. https://doi.org/10.3390/aerospace9020096

Clark, C. (2018). Effects of Aerodynamic Blockage on Stagnation Collection Efficiency in a Wind Tunnel Icing Environment. In 2018 Atmospheric and Space Environments Conference (p. 3832).

2. what is the thermal boundary condition on the wall of the airfoil? Why to write all the flow conditions in the text and also in table 1? Remove the text content. 

3. what are wind tunnel conditions in ref [4]? Are these the same as what the authors are modeling? 

4. Comment on the difference in the modeling and experimental results in figure 3. why are these differences seen?

5. In the case of fig 8 validation, the ice shape is reasonable compared to fig 3. why?

6. In general, work content is a little limited as the authors claim that the primary purpose is to check the limit of blockage ratio for performing an ice wind tunnel test. Since most practical conditions, airfoil AoA is not going to 0. It would be better for authors to report the effect of AoA with various blockage effects.

Author Response

(The authors gave the same response as above.)

Round 2

Reviewer 3 Report

The manuscript can be in the present form